# Neurodynamic Techniques in the Treatment of Mild-to-Moderate Carpal Tunnel Syndrome: A Systematic Review and Meta-Analysis

**DOI:** 10.3390/jcm12154888

**Published:** 2023-07-25

**Authors:** Sheikh Azka Zaheer, Zubair Ahmed

**Affiliations:** 1Institute of Inflammation and Ageing, University of Birmingham, Edgbaston, Birmingham B15 2TT, UK; sheikhazkazaheer92@gmail.com; 2Centre for Trauma Sciences Research, University of Birmingham, Edgbaston, Birmingham B15 2TT, UK

**Keywords:** carpal tunnel syndrome, neurodynamic modulation, physical activity, pain, rehabilitation

## Abstract

Carpal tunnel syndrome (CTS) is a condition that affects the main nerves in the wrist area that causes numbness, tingling, and weakness in the hand and arm. CTS affects 5% of the general population and results in pain in the wrist due to repetitive use, most commonly affecting women and office workers. Conservative management of CTS includes neurodynamic modulation to promote median nerve gliding during upper limb movements to maintain normal function. However, evidence for the benefits of neurodynamic modulation found disparities, and hence, the effectiveness of neurodynamic modulation remains unclear. This study aimed to systematically review the current evidence from randomized controlled trials (RCTs) to establish the effectiveness of neurodynamic techniques as a non-surgical treatment option for CTS. Using the PRISMA guidelines, two authors searched four electronic databases, and studies were included if they conformed to pre-established eligibility criteria. Primary outcome measures included outcomes from the Boston carpal tunnel syndrome questionnaire, while secondary outcomes included nerve conduction velocity, pain, and grip strength. Quality assessment was completed using the Cochrane RoB2 form, and a meta-analysis was performed to assess heterogeneity. Twelve RCTs met our inclusion/exclusion criteria with assessments on 1003 participants in the treatment and control arms. High heterogeneity and some risks of bias were observed between studies, but the results of the meta-analysis showed a significant reduction in our primary outcome, the Boston carpal tunnel syndrome questionnaire-symptom severity scale (mean difference = −1.20, 95% CI [−1.72, −0.67], *p* < 0.00001) and the Boston carpal tunnel syndrome questionnaire-functional severity scale (mean difference = −1.06, 95% CI [−1.53, −0.60], *p* < 0.00001). Secondary outcomes such as sensory and motor conduction velocity increased significantly, while motor latency was significantly reduced, all positively favoring neurodynamic techniques. Pain was also significantly reduced, but grip strength was not significantly different. Our systematic review demonstrates significant benefits of neurodynamic modulation techniques to treat CTS and specifically that it reduces symptom severity, pain, and motor latency, while at the same time improving nerve conduction velocities. Hence, our study demonstrates a clear benefit of neurodynamic techniques to improve recovery CTS.

## 1. Introduction

Carpal tunnel syndrome (CTS) occurs as a result of pressure on the main nerves in the wrist area and is the most common mononeuropathy in the upper limb, accounting for 90% of all peripheral neuropathies [1]. The incidence of CTS is reported to be 3.8–12% with women disproportionately affected than men [1,2,3]. CTS was characterized by sensory and motor symptoms, manifested by numbness, pain, and paresthesia radiating from the wrist to the first three digits. As CTS progresses, muscle atrophy, reduced hand power, and decreased hand dexterity adversely affect the quality of life, activities of daily living, mental well-being, and social participation [4,5,6]. CTS is classified into mild, moderate, and severe types based on electrophysiological diagnosis and different classification systems used [7]. Although CTS was primarily idiopathic, various external and internal risk factors also contribute to the development of CTS. Pregnancy, diabetes, obesity, hypothyroidism, local injection or infection, typing, radial head fractures, and burns increase the risk of developing CTS [8,9,10].

The clinical guidelines recommend conservative management in mild-to-moderate CTS, whereas surgical release is the choice of treatment in severe cases [11]. Conservative management of CTS mainly includes splinting, manual therapy, corticosteroid injections, and orthosis [12,13]. Moreover, it was estimated that 61% of patients with CTS avoid taking surgical options due to post-operative complications and costly surgical procedures, thereby opting for conservative intervention for long-term relief of the symptoms [14]. The existing literature suggested that the annual cost of surgeries for CTS was estimated at around USD 2 billion in the United States with long waiting lists [15]. Implicitly, the huge population is reliant on conservative treatment, and thus, it is essential to evaluate inexpensive treatment options.

Manual therapy facilitates nerve gliding through various techniques including soft tissue and wrist joint mobilization and neurodynamic techniques [16,17]. Neurodynamic modulation (NM) is a revolutionary method for nerve mobilization to manage symptoms in CTS. In general, NM involved treatment regimens consisting of (1) functional massage of the descending part of the trapezius; (2) wrist opening and closing; and (3) gliding and tensioning mobilization of the median nerve (including one-directional proximal and distal glide mobilization and one-directional distal tension mobilization). As the median nerve glides longitudinally during the upper limb movements, NM promotes the normal physiological and mechanical movement of the nerve in CTS [18,19]. It is reported to improve the pressure pain threshold, functional status, and symptom severity in CTS through sliding or gliding maneuvers [20]. Although randomized controlled trials (RCTs) showed beneficial effects of NTs on numbness, pain, and functional status, a recent systematic review found a disparity in its effectiveness for the clinical management of mild-to-moderate CTS [21]. In another systematic review which reported the short-term effects of NM, they were effective in improving pain and function, with very low certainty evidence, but there were no significant effects on symptom severity, distal motor latency, and grip or pinch strength [22].

In addition to this, the superior benefits of NM compared to other manual therapy techniques are controversial. Existing evidence has been restricted to limited comparisons and therapeutic differences while comparing the effectiveness of NM with other conservative treatment options, including kinesio taping, electrotherapy, and manual therapy [23,24,25]. Hence, there is insufficient evidence that highlights an unmet need for the evaluation of this technique’s effectiveness as compared to other manual therapy techniques for improving the symptoms of mild-to-moderate CTS. Therefore, our systematic review aimed to determine the effectiveness of NM, focusing on symptom severity and functional status in mild-to-moderate CTS. We also evaluated if neurodynamic techniques are more effective than other manual therapy techniques.

## 2. Materials and Methods

### 2.1. Literature Search

The relevant studies were identified through searching four electronic databases, i.e., Cinahl, Ovid Medline, Cochrane Central of Registered Trials, and ProQuest from inception to December 2022. The search was completed by S.A.Z. and confirmed by Z.A. Boolean operators AND, NOT, and OR were used. The purpose of using these databases was to extract a wide range of data and prevent biased results. The reference list all included articles were also screened for the extraction of relevant articles meeting the inclusion/exclusion criteria. Our systematic review was based on the Preferred Reporting Items for Systematics Reviews and Meta-Analysis checklist (Appendix A) according to the Cochrane Handbook Guide for the systematic review [26] but has not been registered. Filters applied were as follows: randomized controlled trials (RCTs), English language, and human studies. Conference proceedings, reviews, other systematic reviews, registries, and ongoing trials were excluded.

The MeSh search strategy was used with combinations including carpal tunnel syndrome* AND neurodynamic techniques, carpal tunnel syndrome AND exercise* AND physical activity*, carpal tunnel syndrome AND rehabilitation, and carpal tunnel syndrome AND neurodynamic*. All the results of the specified search from the databases were saved in the Microsoft Excel spreadsheet for the complete record and documentation.

### 2.2. Eligibility Criteria

Two authors (S.A.Z and Z.A) assessed the study title and abstracts according to the PICOS framework for the inclusion criteria of the study: (1) population: adult participants (>18 years of age) diagnosed with mild-to-moderate carpal tunnel syndrome were considered; participants who presented with unilateral or bilateral CTS with symptoms reported up to 6 months were also included; (2) intervention: neurodynamic techniques with or without manual therapy; (3) control: placebo, sham, or any other treatment; (4) primary outcome measure was symptom severity and/or functional status scale of Boston carpal tunnel syndrome questionnaire (BCTQ); (5) secondary outcome measures were nerve conduction velocities (NCV), pain, and grip strength; (6) study design: randomized controlled trials (RCTs) from inception to December 2022; (7) setting: hospitals, orthopedic wards and clinics, rehabilitation centers, and out-patient and in-patient departments.

The exclusion criteria were as follows: non-specified stage of CTS, participants < 18 years of age, co-morbidities (e.g., pregnancy, diabetes, trauma, congenital disorders, obesity, post-surgical release, and hypothyroidism), electrotherapy, corticosteroid injections, plasma-rich protein therapy, kinesio taping, splinting, virtual exercise program, app-based exercises, and nursing home cares.

### 2.3. Study Selection and Data Collection

Two authors (S.A.Z and Z.A) independently screened the abstracts and titles of all the searched data based on predetermined selection criteria. De-duplication of the studies was performed through Endnote 20.4 software, and hand searching was also conducted to reduce the reporting bias. Articles were reviewed and selected for full-text reading that matched our inclusion/exclusion criteria. The process was then checked by the senior author (Z.A), and any discrepancies were resolved through discussion. Studies were excluded from the systematic review based on inappropriate population, intervention, comparison, outcome measure, and study design and setting (PICOS framework).

We extracted study characteristics (study author and year, study design, country, age of participants, gender, study’s inclusion and exclusion criteria, blinding, intervention, outcome measures, and findings of the study) and data for our primary and secondary outcomes (trial author and year, primary outcome and secondary outcome, experimental and control group, post-intervention median and standard deviations, the effect of therapy, timeframe, follow up, and adverse effects). Intervention type, dosage, and frequency of NM with or without manual therapy versus placebo, sham, or standard therapy were retrieved. Data extraction tables were designed according to the Cochrane Consumer and Communication Review Group’s data extraction template in the web-based Google Spreadsheet and Microsoft Word software packages. Four studies were piloted to identify the modifications required in the data extraction table. All of the relevant information was added and stored in the modified versions of the data extraction tables as per the requirement of the eligibility criteria of our study. The trials were re-read and evaluated to extract the important findings, follow-up, frequency of the treatment session, and limitations.

The outcome measures were evaluated by reading the score ranges, baseline characteristics of the participants, and median and standard deviations of the trial’s results. The data of the experimental and control group were compared by identifying trends and differences in the result.

### 2.4. Risk of Bias Assessment

The validity and reliability of each included trial were determined by two authors (S.A.Z) and (Z.A) independently assessing the risk of bias through the Risk of Bias 2 (RoB2) tool by Cochrane recommendations. Disagreements were resolved through discussion among both authors. The overall risk of bias was assessed across five domains including the randomization process, the effect of assignment, missing outcome data, measurement of outcome, and selection of results. The algorithms of ROB 2 were used to analyze the potential risk of bias [27].

### 2.5. Data Synthesis and Statistical Analysis

The data were synthesized qualitatively and quantitatively. We conducted a meta-analysis in Review Manager (RevMan) 5.4 (Cochrane Informatics & Technology, London, UK), using the dichotomous data function employing a random effects model. Mean differences with a 95% confidence interval (CI) were calculated from pre- and post-therapy and used to analyze the effect of the intervention. Cohen’s d or Hedge’s g were not used for data interpretation as meta-analysis in RevMan 5.4 automatically adds weighting to each study result. Assessment of heterogeneity was performed by examining the differences across studies in the meta-analysis and referring to the Q and I^2^ statistics (in percentages). A forest plot was used to graphically represent the results of the meta-analysis.

## 3. Results

### 3.1. Study Selection

A total of 326 studies were retrieved from the Cochrane Central Register of Controlled Trials, Ovid Medline, Proquest, and Cinahl database search. After removal of duplicates, 302 articles were screened according to the inclusion/exclusion criteria for full-text reading. After reviewing titles and abstracts, 265 articles were removed, and out of the remaining 36 studies, the full-text reading excluded 24 studies based on the PICOS framework of this systematic review, including inappropriate population, outcome measure, setting, and intervention. No additional studies were found from reference lists in included studies. Twelve studies were included in our final systematic review [10,14,21,25,28,29,30,31,32,33,34,35], and the PRISMA flow diagram is shown in Figure 1.

### 3.2. Study Characteristics

The 12 included studies in this systematic review were all published between 2000 and 2022 and compared neurodynamic techniques to control treatments or no interventions [10,14,21,25,28,29,30,31,32,33,34,35]. Five studies were multicenter RCTs [14,29,30,31,32], whereas the rest were single-center RCTs [10,21,25,28,33,34,35]. Interestingly, five of the RCTs were from the same first author/authors [14,29,30,31,32], However, these studies tended to include the greatest number of patients per study arm (i.e., 58–78 in each arm). The randomization process varied among all of the RCTs from drawing lots to random number generation, to random sampling [10,14,21,25,28,29,30,31,32,33,34,35]. An overview of the study characteristics can be found in Table 1.

The included studies present results from a total of 1003 patients; 549 received NM, and 454 received sham or control treatments. A further 134 patients, 58 allocated to NM and 76 in the control treatment groups, were enrolled but were excluded from the final analysis for various reasons (Table 2). Both males and females aged 18–85 years with unilateral or bilateral CTS were included in the studies. NM was given over a variety of sessions per week, with some treatments lasting for 4 weeks and others for 10 weeks. In general, neurodynamic techniques were applied by a trained physiotherapist in all of the included studies.

In five of the 12 included RCTs, BCTQ, split into the functional symptom scale (FSS) and symptom severity scale (SSS) were used as the main outcome [14,21,30,31,34]. Two additional studies also used BCTQ, with one study reporting a single value for BCTQ [10], but the other mentioned the use of BCTQ but did not report anything [33]. Nerve conduction velocity studies (NCVs) were performed in five studies [10,14,28,30,31], with three studies breaking these down to sensory conduction velocity (SCV), motor conduction velocity (MCV), and motor latency [14,30,31]. Pain was assessed in eight studies [10,14,21,25,28,30,34,35], using either the visual analogue scale (VAS), the numerical pain rating scale (NPRS), or the West Haven–Yale multidimensional pain inventory (WHYMPI).

Grip strength was assessed in six studies [10,14,21,28,31,35], but once again, one study mentioned baseline grip strength, but no data were reported after the intervention [28]; and finally, disabilities of the arm, shoulder, and hand (DASH [21] and Quick DASH [21,33]) was assessed in three studies. In all included studies, the intervention was either NM or control therapy or no intervention. Other outcome measures included 2-point discrimination (2PD) [29], wrist range of motion [21], ultrasound [35], and the West Haven–Yale multidimensional pain inventory (WHYMPI) [35].

### 3.3. Risk of Bias

The overall risk of bias across all domains assessed was low in only 33.3% of the studies, but 50.0% and 16.6% of studies were judged to contain some concerns to high risk of bias, respectively (Figure 2A,B). While most studies included details of randomization, 16.6% of studies raised some concerns due to the extremely small size of participants (7–15 patients in each group) or it was unclear how randomization was achieved (Figure 2A,B). Missing outcome data, bias due to deviations from the intended interventions and bias in the selection of the reported results were all judged to have some risk of bias. Overall, only two studies had no judged risk of bias [21,28], whereas all of the other studies had some bias, ranging from some concerns to high risk of bias.

### 3.4. Data Synthesis

Overall, of the seven studies that assessed our primary outcome of BCTQ, neurodynamic techniques were effective for the management of symptom severity and functional status in mild-to-moderate CTS (Table 3). For example, BCTQ scores pre- and post-intervention for both SSS and FSS showed the greatest reduction in the NM groups compared to control groups [14,21,30,31,34]. Meta-analysis from the included studies confirmed an overall significant decrease in SSS (mean difference = −1.20, 95% CI [−1.72, −0.67], *p* = 0.00001) (Figure 3A) and FSS (mean difference = −1.06, 95% CI [−1.53, −0.60], *p* < 00001) (Figure 3B) post NM. Inclusion of studies that reported a combined BCTQ to a calculated combined BCTQ in studies that reported separate SSS and FSS (i.e., SSS+FSS) (Table 4) demonstrated a significant decrease in overall BCTQ scores (mean difference −0.89, 95% CI [−1.18—0.60], *p* = 0.00001) (Figure 3C). These results suggest that BCTQ scores, measured immediately after the intervention, were effective in improving outcomes after NM [10,14,21,30,31,34].

Nerve conduction velocities, broken down to SCV (Table 5) and MVC (Table 6) were only provided in three included studies [14,30,31]. Meta-analysis showed that SCV, MCV and motor conduction latency were all significantly improved after NM compared to control treatment (Figure 4).

For example, the mean difference in SCV increased significantly after NM by 12.22 m/s ([95% CI 8.15, 16.28], *p* = 0.00001) (Figure 4A) while the mean difference in MCV also increased significantly after NM by 2.05 m/s ([95% CI 1.46, 4.44], *p* < 0.0001) (Figure 4B). In addition, motor conduction latency (Table 7) was significantly reduced after NM by −0.67 m/s ([95% CI −1.04, −0.30], *p* = 0.0003) (Figure 4C). These results suggest that NM improved overall nerve conduction velocities, increasing sensory and motor conduction velocities while at the same time reducing motor latency.

Results of pain, pre- and post-intervention, were reported in 8 of the 12 studies, using VAS [10,28,34], NPRS [14,21,25,30], or WHYMPI [35] (Table 8). We combined the results from both VAS and NPRS and calculated the difference between pre- and post-interventions (Table 8) and performed a meta-analysis. Our results showed that there was an overall significant reduction in pain by −2.87 (95% CI −4.38, −1.36], *p* = 0.0002) (Figure 5A). However, breaking this down into studies that used VAS to record pain showed that there was an overall reduction in pain, but this did not reach significance (mean difference = −2.64 [95%CI −7.48, 2.20], *p* = 0.29) (Figure 5B). Studies that used NPRS to assess pain showed an overall significant reduction in pain, by −2.65 ([95%CI −4.03, −1.28], *p* = 0.0002) (Figure 5C). Finally, meta-analysis of grip strength, which was reported in five studies (Table 9) showed that NM had no significant effect when compared to controls (mean difference = −0.02, [95% CI −1.31, 1.27], *p* = 0.97) (Figure 6). These results demonstrate that while NM can significantly reduce pain after treatment, grip strength remains unaffected.

## 4. Discussion

In this study, we evaluated the effectiveness of NM (gliding and sliding maneuvers) in reducing symptom severity and improving the functional status in adults with mild-to-moderate CTS. Overall, 12 RCTs met our inclusion/exclusion criteria with results for 1003 participants, including 549 of those treated by NM presented in the RCTs [10,14,21,25,28,29,30,31,32,33,34,35]. Our systematic review showed that BCTQ was significantly improved after NM in patients with CTS, as was sensory and motor conduction velocities with significantly reduced motor latencies. Pain was also significantly reduced, but grip strength remained unaffected. These results support the use of NM for effective management of CTS patients. However, since some of the studies were judged to possess some risk bias, improved quality, well-controlled RCTs with larger numbers of patients could improve the quality of the evidence in support of NM for the treatment of CTS.

All of the included studies suggested that NM was effective in improving symptom severity and functional status in mild-to-moderate CTS, assessed using the BCTQ [10,14,21,30,31,33,34]. However, as with other systematic reviews on NM and CTS clinical management, there were inconsistent methodological issues found in the RCTs included in our study, with studies using a variety of measures for the same thing. For example, some studies broke down BCTQ into BCTQ-SSS and BCTQ-FSS [14,21,30,31,34], while other studies reported only one value for BCTQ [10,33], making the interpretation of the results more difficult. Nerve conduction was also assessed differently with some studies breaking this down to SCV, MCV, and motor latency [14,30,31], while other studies only reported a single nerve conduction value [10,28]. Pain was also assessed using either VAS [10,25,28,34], NPRS [14,21,25,30], or WHYMPI [35], again making the interpretation of the outcome of NM more difficult to compare across studies. Even DASH was assessed by either QuickDASH [21,33] or DASH [28]. Thus, methodological issues in RCTs could weaken the reliability and confidence in the results as studies could be subjected to under- or over-estimation of findings.

One way we reconciled these differences in this systematic review was to calculate the difference pre- and post-NM and use these values to compare the results across the studies. As a precaution, we also assessed individual methods separately. In general, there was good agreement between the individually assessed methods and the combined results except for pain where the use of VAS resulted in a non-significant change by NM, but when combined with NPRS results, there was an overall significant effect. We recommend that future studies are designed with care to trial a design such that a standardized set of measures are assessed in CTS and any potential therapy. Our study shows that BCTQ, broken down into BCTQ-SSS and BCTQ-FSS, nerve conduction studies to assess SCV, MCV, and motor latency, as well as measurement of pain using NPRS might be the most sensitive measures in CTS.

Although other systematic reviews have been performed to evaluate the effectiveness of NM on function in CTS patients, our study evaluated the greatest number of outcomes and found significant differences where others showed no benefits of NM. For example, one systematic review reported no significant effects on symptom severity, distal motor latency, and grip and pinch strength but with low certainty [21]. Another study reported that NM was superior to no treatment on pain and BCTQ but with low quality evidence, while NM did not demonstrate clinical effectiveness [36]. Another study of reported on 13 clinical trials which showed improvements in pain, pressure, and function of CTS after NM, but when compared to other therapies, only two studies reported better results from standard of care, and three studies reported greater and earlier pain relief and function after NM techniques than when compared to conservative techniques [37]. However, most of the studies were deemed low quality [37].

We also found differences in the risk of bias within RCTs, with only 2 of the 12 studies presenting a low risk of bias. The highest risks of bias were judged to be in the treatment of missing outcome data, bias in the measurement of the outcome, bias in the selection of reported results, and bias arising from the randomization process. This means that our study data, despite showing statistically significant benefits of NM in the management of CTS, must be interpreted with caution. Because clinical guidelines play an integral role in clinical decision making [38], the low quality of evidence could be a potential reason why recent clinical guidelines do not, as yet, recommend NM as a conservative treatment option in mild-to-moderate CTS [39]. Our meta-analysis, however, has highlighted significant differences in improvements in different parameters assessed after the application of NM and suggests that further high quality RCTs are required to reach a definitive decision in the management of CTS.

Moreover, NM apparently generated biomechanical, physiological, and neuroimmune responses, but the exact mechanism differed greatly in human and animal studies. For example, NM decreased intra-neural oedema, activated analgesic neuronal pathways, induced anti-inflammatory changes, and desensitized mechanical compression by enhancing the median nerve excursion and nerve diameter [40]. Additionally, a recent systematic review of animal studies reported that NM predisposes a beneficial impact through modulating neurotrophins, neuroinflammation, and opioid systems, leading to decreased mechanical hyperalgesia [40]. In addition, the application of NM on cadavers revealed physiological mechanisms of pain relief, concluding that it enhanced the intraneural dispersion as well as decreasing intra-neural oedema surrounding the tissues [41]. Therefore, the exact mechanism of NM on symptoms and functional status in mild-to-moderate CTS remains unclear.

In the included RCTs, the application of NM varied in dosage, type, and length of treatment. Categorically, gliding enhanced the gliding motion by increasing the length of the nerve bed, whereas the sliding technique released the tension around the median nerve [42]. Studies have also suggested that the sliding technique was more beneficial for CTS [42]. Similarly, to another systematic review [22], we also found inconsistencies in the implementation of NM. Five studies used the same NM technique, dose, and duration of treatment since all of these studies were from the same first author [14,29,30,31,32]. This presented a potential issue in that there could have been a high risk of bias as it is the same first author; however, to their credit, these studies were multicenter and had the highest numbers of patients. This mitigated some of the initial concerns over these studies.

Other studies used a variety of approaches, with treatment ranging from 60 min weekly to several sessions per week over 3–4 weeks [10,21,28,33,34,35]. Thus, the inconsistency and variation in dosage of NM not only could have contributed to the overall outcomes, but it also presents a challenging decision for the physical therapist and clinician to select the intervention for achieving optimal outcomes.

The long-lasting effect of NM is also unknown, and several factors may play a key role in determining the duration of NM in mild-to-moderate CTS. We assume that NM imposes short-term relief of symptoms as post-intervention effects in the included RCTs were assessed immediately after the treatment [10,14,21,25,28,29,30,31,32,33,34,35]. In two studies, a combination of manual therapy and electrotherapy (TENS or ultrasound) was administered along with NM, and this could explain the potential influence on some of the results in these studies [29,30,33]. In agreement with our observations, Barrio et al. [43] also found that there was a lack of follow-up data, due to which the lasting impact of NM could not be established. However, Fernandez-des-Las-Penas et al. [44] reported in a 4-year follow-up study that manual therapy, as well as NM, had similar effectiveness as compared to surgery, with only 15% of women requiring surgical treatment after manual therapy with NM. Hence, the addition of follow-up data would have been useful to establish the long-term effectiveness of NM in recovery from mild-to-moderate CTS.

### 4.1. Limitations

Our study has several limitations. Despite there being a reasonable number of RCTs, the total number of patients included was only 549 with CTS, restricting the generalizability to larger populations. The Cochrane handbook suggests that the wide confidence interval could have resulted due to the high heterogeneity in our included RCTs [45]. We suspected that the high heterogeneity found in both outcome measures might influence the estimation of the effect size. Indeed, it was indicated that the existing RCTs were designed with non-homogenous data, and the concerns with the risk of bias in included studies may limit the value of this systematic review, as the interpretations from our study must be taken with caution.

Secondly, we could not perform a meta-analysis of all of the included RCTs due to the unavailability of homogeneous numerical data for our chosen outcomes across all studies. This made it challenging to interpret the effect of NM on mild-to-moderate CTS. Thirdly, there might be a language or publication bias because we only analyzed studies in the English language, and non-English articles were excluded as per the eligibility criteria.

### 4.2. Future Directions

This systematic review implies that the methodological quality of the RCTs should be improved through robust methodology with numerical data analysis. In future studies, multicenter, large-sample-size, and homogenous data collection across defined assessments in RCTs should be considered to definitively determine the value of NM in the treatment of CTS. Additionally, it would be interesting to find out the duration of the effectiveness of NM in CTS by including a longer-term follow-up in RCTs.

Furthermore, as we have discussed, various underlying pathophysiological mechanisms were generated in response to NM, so future research must quantify, validate, and analyze NM in association with outcome measures. A similar study illustrated that the sciatic nerve movement during neural mobilization could be visualized and quantified [44]. So, in mild-to-moderate CTS, relevant comparisons to assess the effectiveness of each subjective or functional symptom (pain, numbness, muscle strength, ROM, etc.) either as neurophysiological or mechanical change stimulated with the application of NM could be developed in vivo. Eventually, this study design could be helpful to determine which therapeutic mechanisms may be activated by NM.

## 5. Conclusions

This systematic review evaluated the therapeutic effectiveness of NM on symptom severity and functional status in mild-to-moderate CTS. Our findings demonstrated significant benefits on several parameters in CTS but found several disparities in methodologies and high heterogeneity, while only 12 RCTs met our inclusion/exclusion criteria, and there were only 549 study participants in the NM group. There is potential for future trials to develop a more robust methodology with homogenous data collection and a much larger cohort to accurately evaluate the effectiveness of NM in the recovery from mild-to-moderate CTS.

## Figures and Tables

**Figure 1 jcm-12-04888-f001:**
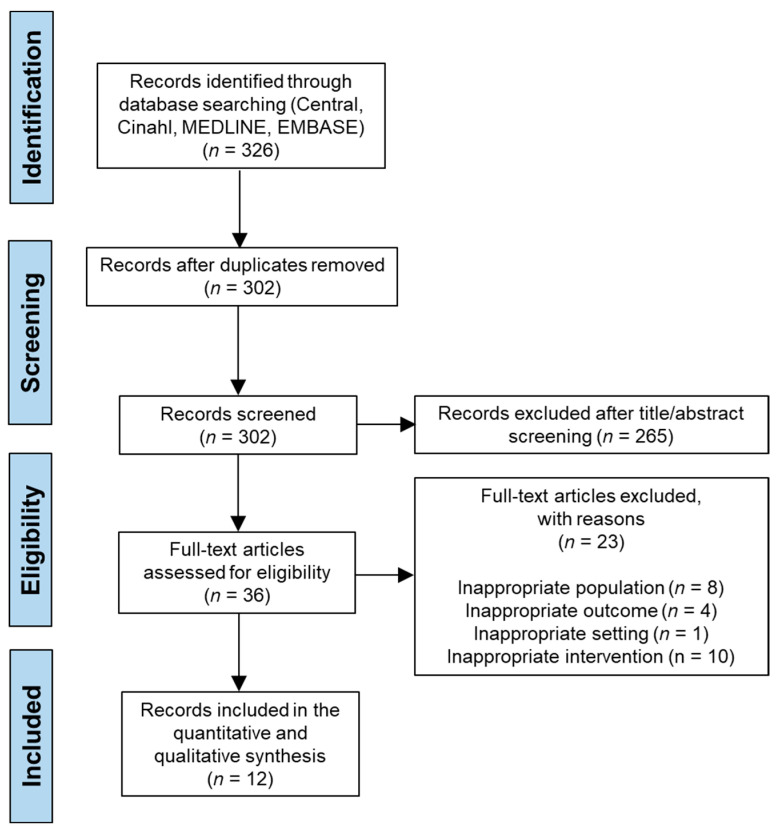
PRISMA flow chart.

**Figure 2 jcm-12-04888-f002:**
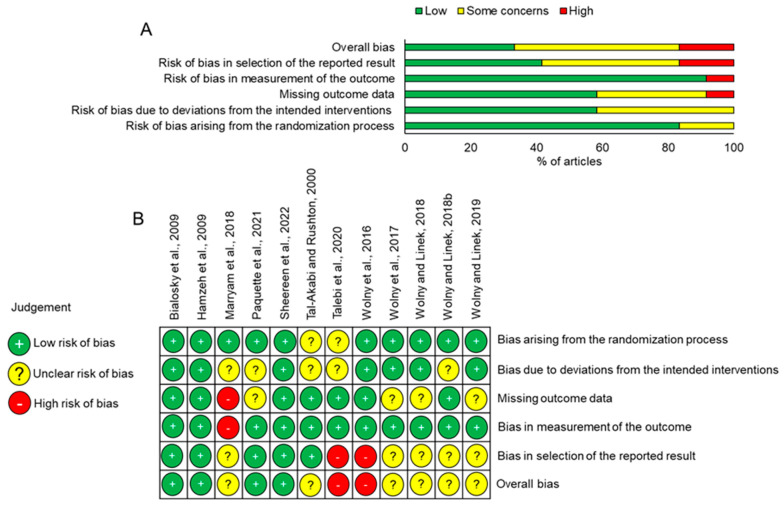
Risk of bias analysis in included studies. (**A**) Risk of bias summary in all studies. (**B**) Risk of bias in each study against the 5 domains [27].

**Figure 3 jcm-12-04888-f003:**
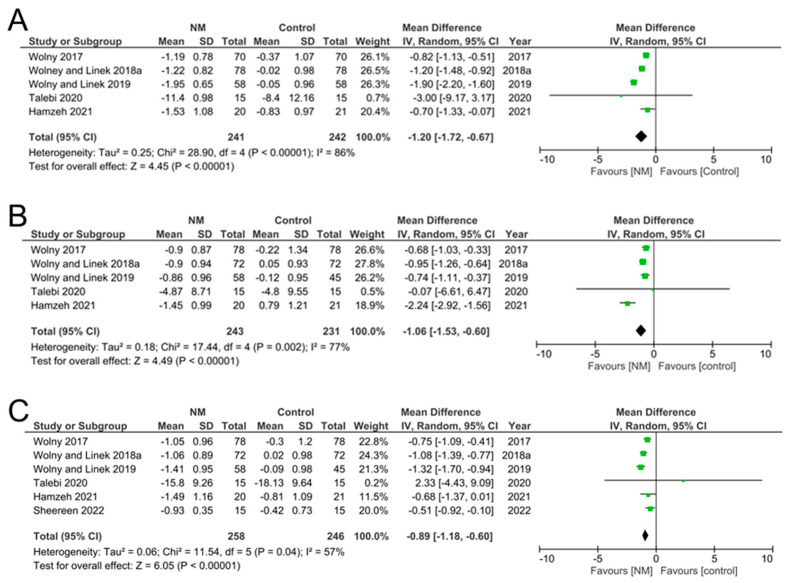
Meta-analysis results of BCTQ in included studies. (**A**) Meta-analysis of BCTQ-SSS (symptom severity scale). (**B**) Meta-analysis of BCTQ-FSS (functional status scale). (**C**) Meta-analysis of combined SSS and FSS in included studies. Results reflected significant improvements in the NM-treated groups, reflected by lower scores, compared to controls.

**Figure 4 jcm-12-04888-f004:**
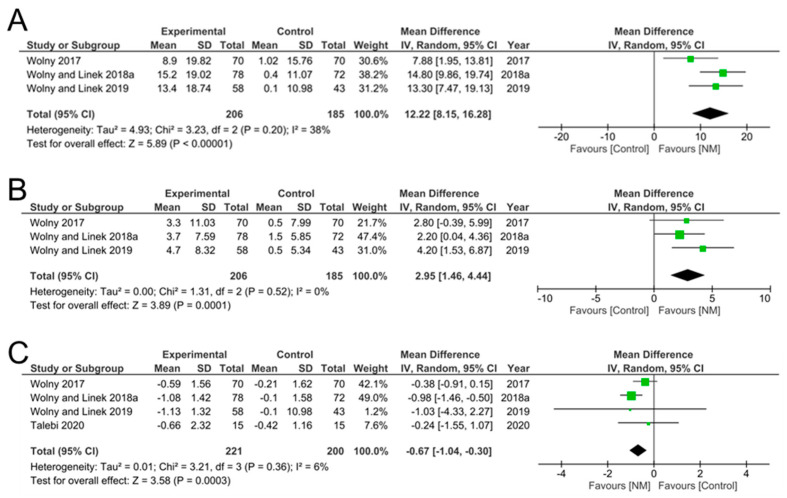
Meta analysis of nerve conduction studies. (**A**) SCV (sensory conduction velocity) and (**B**) MVC (motor conduction velocity) increased significantly, while (**C**) motor latency was significantly reduced by NM.

**Figure 5 jcm-12-04888-f005:**
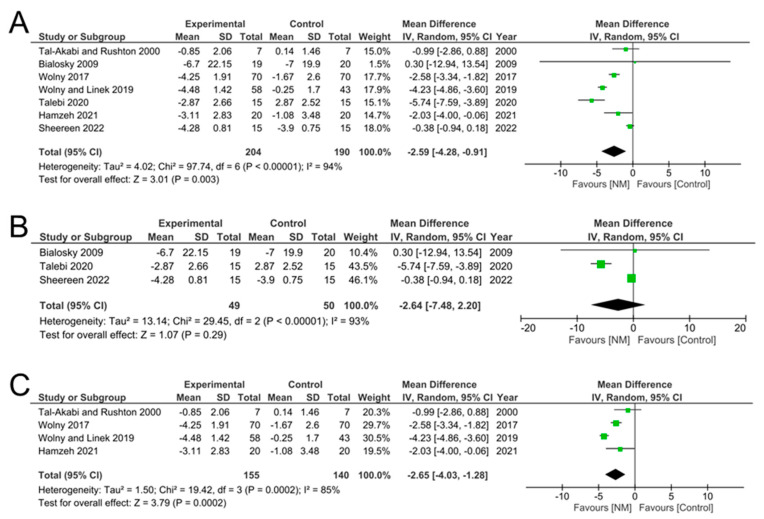
Meta-analysis of pain scores showed a significant reduction in pain after NM compared to control. (**A**) Data from all studies combined into the meta-analysis. Individual data from studies using the (**B**) visual analogue scale or the (**C**) numerical pain rating scale.

**Figure 6 jcm-12-04888-f006:**
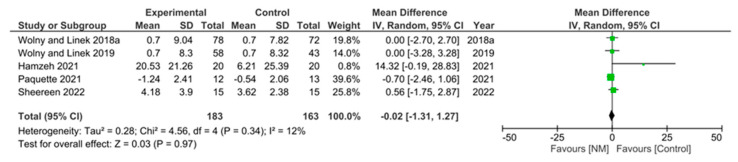
Meta-analysis of grip strength demonstrated no significant improvements in the NM-treated group.

**Table 1 jcm-12-04888-t001:** Characteristics of the included studies.

Study	Study Year	Country	Age (Years); Gender	No. of Participants	Study Inclusion Criteria	Outcome Measure	Intervention	Frequency of Interventions
Tal-Akabi and Rushton. [25]	2000	UK	29–85; male and female	21 (7 NM; 7 CBM; 7 Control)	CTS diagnosis by physician: positive Phalen’s test, positive Tinel’s sign, and positive electrodiagnostic test.	NPRS FBS ULTT2a	Neurodynamic modulation Carpal bone mobilization (CBM) No intervention	Not stated
Bialosky et al. [28]	2009	USA	18–70; male and female	40 (19 NM; 20 Controls)	Presence of CTS as defined by pain or paresthesia in the median nerve distribution and/or clinical examination findings consistent with CTS. CTS was present for greater than 12 weeks with pain rating of 4/10.	MVAS DASH Grip strength NCS	Neurodynamic technique “directly stresses the median nerve through shoulder, elbow, and wrist movements” or “indirectly stresses the median nerve through shoulder, elbow, and wrist movements” Sham—no stress across median nerve	2 session per week over 3 weeks
Wolny et al. [29]	2016	Poland	26–72; male and female	210: (140 NM; 70 Control)	CTS diagnosis by physician: numbness and tingling of the median nerve, paresthesia, positive Phalen’s test, positive Tinel’s sign, and pain in wrist radiating to shoulder	2PD	Neuromodulation: manual therapy Electrophysical modalities (laser and ultrasound)	20 sessions over 10 weeks
Wolny et al. [30]	2017	Poland	>18; male and female	140 (70 NM; 70 Control)	Numbness and tingling; nighttime paresthesia; positive Phalen test and Tinel sign Pain radiating to the shoulder NCS diminished nerve conduction values, increases motor latency	NCS BCTQ NPRS	Manual therapy group: neurodynamic techniques and carpal bone mobilizations Electrotherapy group: Laser and Ultrasound	20 sessions over 10 weeks
Wolny and Linek [31]	2018a	Poland	26–72; male and female	150 (78 NM; 72 Control)	CTS diagnosed based on history, clinical examination: numbness and tingling of the median nerve, paresthesia, positive Phalen’s test, positive Tinel’s sign, and pain in wrist radiating to shoulder and NCS	BCTQ NCS Grip strength Pinch strength	Neuromodulation: neurodynamic techniques Sham glide proximal mobilization	20 sessions over 10 weeks
Wolny and Linek [32]	2018b	Poland	26–72; male and female	189 (102 NM; 87 Control)	CTS diagnosed based on history, clinical examination: numbness and tingling of the median nerve, paresthesia, positive Phalen’s test, positive Tinel’s sign, and pain in wrist radiating to shoulder and NCS	SF-36 PF RF BP GH	Manual therapy including neurodynamic techniques Control group: no therapy	20 sessions over 10 weeks
Maryam et al. [33]	2018	Pakistan	25–55; male and female	27 (13 NM; 14 Control)	Clinically diagnosed mild-to-moderate CTS	BCTQ QuickDASH	Control group: nerve tendon gliding exercises with electrotherapy (TENS, Ultrasound) Experimental group: Neurodynamic techniques with electrotherapy (TENS and Ultrasound)	3 sessions per week over 4 weeks
Wolny and Linek [14]	2019	Poland	53.85 ± 9.60	103 (58 NM; 43 Control)	CTS diagnosed based on history, clinical examination: numbness and tingling of the median nerve, paresthesia, positive Phalen’s test, positive Tinel’s sign, and pain in wrist radiating to shoulder and NCS	BCTQ NPRS NCS Grip strength	Neurodynamic techniques No treatment	20 sessions over 10 weeks
Talebi et al. [34]	2020	Iran	30–50; male and female	30 (15 NM; 15 Control)	Positive findings in the clinical examination (complains of pain, numbness or tingling in the first three digits for 6 months, positive Phalen’s sign) and electro-diagnostic findings.	BCTQ VAS Distal latency of median nerve	Mechanical interface group Nerve mobilization group	3 sessions per week over 4 weeks
Hamzeh et al. [21]	2021	Jordan	>18; male and female	41 (20 NM; 21 Control)	CTS diagnosed based on history, clinical examination: numbness and tingling of the median nerve, paresthesia, positive Phalen’s test, presence of flick sign, nerve conduction <50 m/s, and/or increased latency >4 m/s.	BCTQ QuickDASH NPRS Wrist ROM Grip strength	Neurodynamic techniques Control group (exercise therapy)	60 min weekly session for 4 weeks
Paquette et al. [35]	2021	Canada	18–70; male and female	30 (12 NM; 13 Control)	CTS diagnosis, confirmed by electrodiagnostic test	Ultrasound WHYMPI DASH Grip strength	Neurodynamic techniques Control group (no interventions)	45 repetitions/day for 4 weeks
Sheereen et al. [10]	2022	India	30–59; male and female	30 (15 NM; 15 Control)	Pain, tingling, or paresthesia in the hand including thumb, index finger, middle finger and radial half of the ring finger, VAS of 4–7, positive Tinel’s sign and Phalen’s test, sleep disturbance caused by hand pain, positive nerve conduction study for distal motor latency of >4.4 m/s.	BCTQ VAS NCS Grip strength	Neurodynamic techniques Control group (carpal bone mobilization)	3 alternate days for 3 weeks

Notes: BCTQ = Boston carpal tunnel questionnaire; CBM = carpal bone mobilization; CTS = carpal tunnel syndrome; DASH = disabilities of arm, shoulder, and hand; FBS = functional box scale; MVAS = mechanical visual analogue scale; NCS = nerve conduction studies; NPRS: numerical pain rating scale; 2PD = two-point discrimination scale; NM = neurodynamic modulation; ROM = range of motion; ULTT2a = upper limb tension test 2a; VAS = visual analogue scale; WHYMPI = West Haven–Yale multidimensional pain inventory.

**Table 2 jcm-12-04888-t002:** Patients enrolled but excluded from the final analysis, along with reasons for exclusion.

Study	Number of Patients Excluded from Final Analysis	Reason for Exclusion
Tal-Akabi and Rushton, 2000 [25]	NM group, *n* = 0 Control group, *n* = 0	N/A
Bialosky et al., 2009 [28]	NM group, *n* = 1 Control group, *n* = 0	Did not respond or keep up follow up appointments.
Wolny et al., 2016 [29]	NM group, *n* = 10 Control group, *n* = 10	6 lacked final results of nerve conduction, 2 did not complete final examination form, 2 had comorbidities that resulted in exclusion. 5 lacked final nerve conduction results, 3 resigned from the experiment, 2 had other reasons for withdrawal.
Wolny et al., 2017 [30]	NM group, *n* = 10 Control group, *n* = 10	6 lacked final results of nerve conduction, 2 did not complete final examination form, 2 had comorbidities that resulted in exclusion. 5 lacked final nerve conduction results, 3 resigned from the experiment, 2 had other reasons for withdrawal.
Wolny and Linek, 2018a [31]	NM group, *n* = 12 Control group, *n* = 18	2 resigned, 7 lacked final results of nerve conduction, 3 had other diseases as comorbidities. 4 resigned, 10 lacked final results of nerve conduction, 4 had other diseases as comorbidities.
Wolny and Linek, 2018b [32]	NM group, *n* = 10 Control group, *n* = 25	3 resigned, 4 lacked final nerve conduction results, 3 had other diseases as comorbidities. 12 resigned, 8 lacked final nerve conduction results, 5 had other diseases as comorbidities.
Marryam et al., 2018 [33]	NM group, *n* = 2 Control group, *n* = 3	Did not complete follow up examinations.
Wolny and Linek, 2019 [14]	NM group, *n* = 2 Control group, *n* = 10	Lacked final nerve conduction results. 6 lacked final nerve conduction results, 4 had other diseases as comorbidities.
Talebi et al., 2020 [34]	NM group, *n* = 5 Control group, *n* = 6	5 lost to follow-up. 4 lost to follow-up, 2 resigned for personal reasons.
Hamzeh et al., 2021 [21]	NM group, *n* = 6 Control group, *n* = 4	2 lost contact, 1 resigned, 1 had stroke, 1 left the country and 1 had an additional hip fracture. 4 lost contact.
Paquette et al., 2021 [35]	NM group, *n* = 0 Control group, *n* = 0	N/A
Sheereen et al., 2022 [10]	NM group, *n* = 0 Control group, *n* = 0	N/A

**Table 3 jcm-12-04888-t003:** Computed results from studies assessing BCTQ.

Study	Groups	Pre-Intervention (Mean ± SD)	Post-Intervention (Mean ± SD)	Difference (± SD)
Wolny et al., 2017 [30]	NM group, *n* = 70 Control group, *n* = 70	2.89 ± 0.79 2.86 ± 0.84	1.84 ± 0.55 2.56 ± 0.86	−1.05 ± 0.96 −0.30 ± 1.20
Wolny and Linek, 2018a [31]	NM group, *n* = 78 Control group, *n* = 72	2.92 ± 0.70 2.96 ± 0.68	1.86 ± 0.55 2.98 ± 0.70	−1.06 ± 0.89 0.02 ± 0.98
Wolny and Linek, 2019 [14]	NM group, *n* = 58 Control group, *n* = 45	2.93 ± 0.68 2.96 ± 0.69	1.52 ± 0.66 2.87 ± 0.70	−1.41 ± 0.95 −0.09 ± 0.98
Talebi et al., 2020 [34]	NM group, *n* = 15 Control group, *n* = 15	23.93 ± 7.30 24.73 ± 8.50	8.13 ± 5.70 6.6 ± 4.54	−15.80 ± 9.26 −18.13 ± 9.64
Hamzeh et al., 2021 [21]	NM group, *n* = 20 Control group, *n* = 21	2.99 ± 0.87 2.67 ± 0.80	1.50 ± 0.77 1.86 ± 0.74	−1.49 ± 1.16 −0.81 ± 1.09
Shereen et al., 2022 [10]	NM group, *n* = 15 Control group, *n* = 15	2.28 ± 0.32 2.30 ± 0.47	1.35 ± 0.15 1.88 ± 0.56	−0.93 ± 0.35 −0.42 ± 0.73

Notes: NM = neurodynamic modulation.

**Table 4 jcm-12-04888-t004:** Quantitative results from studies assessing BCTQ.

Study	Primary Outcome	Groups	Pre-Intervention (Mean ± SD)	Post-Intervention (Mean ± SD)	Difference (± SD)
Wolny et al., 2017 [30]	BCTQ-SSS	NM group, *n* = 70	2.97 ± 0.63	1.78 ± 0.47	−1.19 ± 0.78
		Control group, *n* = 70	2.94 ± 0.74	2.57 ± 0.77	−0.37 ± 1.07
	BCTQ-FSS	NM group, *n* = 70	2.80 ± 0.94	1.90 ± 0.62	−0.9 ± 0.87
		Control group, *n* = 70	2.77 ± 0.94	2.55 ± 0.95	−0.22 ± 1.34
Wolny and Linek, 2018a [31]	BCTQ-SSS	NM group, *n* = 78	2.99 ± 0.67	1.77 ± 0.48	−1.22 ± 0.82
		Control group, *n* = 72	2.88 ± 0.72	2.86 ± 0.72	−0.02 ± 0.98
	BCTQ-FSS	NM group, *n* = 78	2.84 ± 0.72	1.94 ± 0.61	−0.9 ± 0.94
		Control group, *n* = 72	3.04 ± 0.64	3.09 ± 0.68	0.05 ± 0.93
Wolny and Linek, 2019 [14]	BCTQ-SSS	NM group, *n* = 58	3.03 ± 0.65	1.08 ± 0.68	−1.95 ± 0.65
		Control group, *n* = 45	2.92 ± 0.71	2.87 ± 0.68	−0.05 ± 0.96
	BCTQ-FSS	NM group, *n* = 58	2.82 ± 0.71	1.96 ± 0.64	−0.86 ± 0.96
		Control group, *n* = 45	2.99 ± 0.67	2.87 ± 0.71	−0.12 ± 0.95
Talebi et al., 2020 [34]	BCTQ-SSS	NM group, *n* = 15	30.66 ± 7.82	19.26 ± 5.48	−11.40 ± 0.98
		Control group, *n* = 15	30.13 ± 8.95	21.73 ± 8.22	−8.40 ± 12.16
	BCTQ-FSS	NM group, *n* = 15	17.20 ± 6.77	12.33 ± 5.48	−4.87 ± 8.71
		Control group, *n* = 15	19.33 ± 8.05	14.53 ± 5.13	−4.8 ± 9.55
Hamzeh et al., 2021 [21]	BCTQ-SSS	NM group, *n* = 20	3.17 ± 0.86	1.64 ± 0.66	−1.53 ± 1.08
		Control group, *n* = 21	2.71 ± 0.76	1.88 ± 0.60	−0.83 ± 0.97
	BCTQ-FSS	NM group, *n* = 20	2.80 ± 0.87	1.35 ± 0.48	−1.45 ± 0.99
		Control group, *n* = 21	2.63 ± 0.84	1.84 ± 0.87	−0.79 ± 1.21
Shereen et al., 2022 [10]	BCTQ	NM group, *n* = 15	2.28 ± 0.32	1.35 ± 0.15	−0.93 ± 0.35
		Control group, *n* = 15	2.3 ± 0.47	1.88 ± 0.56	−0.42 ± 0.73

Notes: BCTQ = Boston carpal tunnel questionnaire; SSS = symptom severity scale; FSS = functional status scale.

**Table 5 jcm-12-04888-t005:** Nerve conduction studies—SCV.

Study	Groups	Pre-Intervention (Mean ± SD)	Post-Intervention (Mean ± SD)	Difference (± SD)
Wolny et al., 2017 [30]	NM group, *n* = 70 Control group, *n* = 70	26.20 ± 15.70 38.20 ± 11.10	35.10 ± 12.10 39.22 ± 11.19	8.90 ± 19.82 1.02 ± 15.76
Wolny and Linek, 2018a [31]	NM group, *n* = 78 Control group, *n* = 72	24.60 ± 15.30 24.70 ± 7.89	39.80 ± 11.30 25.10 ± 7.77	15.20 ± 19.02 0.40 ± 11.07
Wolny and Linek, 2019 [14]	NM group, *n* = 58 Control group, *n* = 43	24.9 ± 15.1 25.8 ± 7.81	38.3 ± 11.1 25.90 ± 7.72	13.40 ± 18.74 0.10 ± 10.98

**Table 6 jcm-12-04888-t006:** Nerve conduction studies—MCV.

Study	Groups	Pre-Intervention (Mean ± SD)	Post-Intervention (Mean ± SD)	Difference (± SD)
Wolny et al., 2017 [30]	NM group, *n* = 70 Control group, *n* = 70	53.2 ± 7.80 54.8 ± 5.6	56.5 ± 7.8 55.3 ± 5.7	3.30 ± 11.03 0.50 ± 7.99
Wolny and Linek, 2018a [31]	NM group, *n* = 78 Control group, *n* = 72	52.4 ± 3.53 52.6 ± 3.94	56.1 ± 6.72 54.1 ± 4.32	3.70 ± 7.59 1.50 ± 5.85
Wolny and Linek, 2019 [14]	NM group, *n* = 58 Control group, *n* = 43	51.10 ± 5.15 53.1 ± 3.44	55.8 ± 6.92 53.6 ± 4.08	4.70 ± 8.32 0.50 ± 5.34

**Table 7 jcm-12-04888-t007:** Motor conduction latency.

Study	Groups	Pre-Intervention (Mean ± SD)	Post-Intervention (Mean ± SD)	Difference (± SD)
Wolny et al., 2017 [30]	NM group, *n* = 70 Control group, *n* = 70	5.61 ± 1.08 5.45 ± 1.12	5.02 ± 1.13 5.24 ± 1.17	−0.59 ± 1.56 −0.21 ± 1.62
Wolny and Linek, 2018a [31]	NM group, *n* = 78 Control group, *n* = 72	5.51 ± 1.08 5.43 ± 1.11	4.43 ± 0.81 5.33 ± 1.13	−1.08 ± 1.42 −0.10 ± 1.58
Wolny and Linek, 2019 [14]	NM group, *n* = 58 Control group, *n* = 43	5.62 ± 1.11 5.51 ± 1.17	4.49 ± 0.72 5.41 ± 1.18	−1.13 ± 1.32 −0.10 ± 1.67
Talebi et al., 2020 [34]	NM group, *n* = 15 Control group, *n* = 15	6.26 ± 1.85 6.18 ± 1.65	5.60 ± 1.40 5.76 ± 1.15	−0.66 ± 2.32 −0.42 ± 1.16

Notes: NM = neurodynamic modulation.

**Table 8 jcm-12-04888-t008:** Pain assessment using VAS or NPRS.

Study	Groups	Pre-Intervention (Mean ± SD)	Post-Intervention (Mean ± SD)	Difference (± SD)
Tal-Akabi and Rushton, 2000 [25]	NM group, *n* = 7 Control group, *n* = 7	2.42 ± 1.51 2.00 ± 1.29	1.57 ± 1.40 2.14 ± 0.69	−0.85 ± 2.06 0.14 ± 1.46
Bialosky et al., 2009 [28]	NM group, *n* = 19 Control group, *n* = 20	22.70 ± 16.30 14.90 ± 15.80	16.0 ± 15.0 7.90 ±12.10	−6.70 ± 22.15 −7.00 ± 19.90
Wolny et al., 2017 [30]	NM group, *n* = 70 Control group, *n* = 70	5.72 ± 1.49 5.25 ± 1.75	1.47 ± 1.20 3.58 ± 1.93	−4.25 ± 1.91 −1.67 ± 2.60
Wolny and Linek, 2019 [14]	NM group, *n* = 58 Control group, *n* = 43	5.86 ± 1.46 5.71 ± 1.34	1.38 ± 1.01 5.46 ± 1.05	−4.48 ± 1.42 −0.25 ± 1.70
Talebi et al., 2020 [34]	NM group, *n* = 15 Control group, *n* = 15	6.40 ± 1.45 6.80 ± 1.65	3.53 ± 2.23 3.93 ± 1.90	−2.87 ± 2.66 −2.87 ± 2.52
Hamzeh et al., 2021 [21]	NM group, *n* = 20 Control group, *n* = 21	4.17 ± 2.23 3.17 ± 2.49	1.06 ± 1.75 2.09 ± 2.43	−3.11 ± 2.83 −1.08 ± 3.48
Sheereen et al., 2022 [10]	NM group, *n* = 15 Control group, *n* = 15	6.30 ± 0.65 6.20 ± 0.47	2.02 ± 0.49 2.30 ± 0.58	−4.28 ± 0.81 −3.90 ± 0.75

Notes: NM = neurodynamic modulation.

**Table 9 jcm-12-04888-t009:** Grip strength (cylindrical grip) (kg).

Study	Groups	Pre-Intervention (Mean ± SD)	Post-Intervention (Mean ± SD)	Difference (± SD)
Wolny and Linek, 2018a [31]	NM group, *n* = 78 Control group, *n* = 72	27.7 ± 6.66 29.6 ± 5.67	28.4 ± 6.11 30.3 ± 5.38	0.70 ± 9.04 0.70 ± 7.82
Wolny and Linek, 2019 [14]	NM group, *n* = 58 Control group, *n* = 43	28.10 ± 6.11 29.4 ± 6.02	28.80 ± 5.62 30.10 ± 5.74	0.70 ± 8.30 0.70 ± 8.32
Hamzeh et al., 2021 [21]	NM group, *n* = 20 Control group, *n* = 21	24.88 ± 16.59 23.43 ± 17.21	35.41 ± 13.30 29.64 ± 18.67	20.53 ± 21.26 6.21 ± 25.39
Paquette et al., 2021 [35]	NM group, *n* = 12 Control group, *n* = 13	4.24 ± 1.98 3.89 ± 1.71	3.00 ± 1.38 3.35 ± 1.14	−1.24 ± 2.41 −0.54 ± 2.06
Sheereen et al., 2022 [10]	NM group, *n* = 15 Control group, *n* = 15	17.08 ± 2.02 17.42 ± 1.20	21.26 ± 3.34 21.04 ± 2.06	4.18 ± 3.90 3.62 ± 2.38

Notes: NM = neurodynamic modulation.

## Data Availability

The study data are available from the corresponding author upon reasonable request.

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
