# Peer review of "Neurodynamic Techniques in the Treatment of Mild-to-Moderate Carpal Tunnel Syndrome: A Systematic Review and Meta-Analysis"

_jcm, 2023, doi:10.3390/jcm12154888_

Round 1
Reviewer 1 Report
First of all, I want to say that it has been a pleasure reviewing your manuscript. I think this is an interesting topic for physicians managing this prevalent condition. In short, you need to define the pathology in the first line. Also in the abstract, the term systematic review is repeated several times in the last lines.There are no coments.
Author Response
Comment 1: First of all, I want to say that it has been a pleasure reviewing your manuscript. I think this is an interesting topic for physicians managing this prevalent condition.
Author response: Thank you for your kind comments. We are very grateful for your time in reviewing our manuscript.
Comment 2: In short, you need to define the pathology in the first line. Also in the abstract, the term systematic review is repeated several times in the last lines.
Author response: We have added a definition of CTS in the first line of the Abstract and the Introduction and have replaced the use of the word systematic review with study.
Reviewer 2 Report
This study performed a systematic review and meta-Analysis of the effectiveness of neurodynamic techniques on mild to moderated carpal tunnel syndrome (CTS). The authors searched the randomized controlled trials from the Cinahl, Ovid Medline, Cochrane Central of Registered Trials, and ProQuest from inception to December 2022 and used PRISMA guidelines to build the review of the included 12 clinical trials. The manuscript is well written, but there are some concerns that should be addressed.
1. Although the authors made some discussion regarding the discrepancies of neuromodulation (NM) implemented in the included 12 studies, it is recommended to make a definition of NM technique and explain their applying methods (e.g., gliding and sliding maneuvers) in the Introduction section.
2. Lines 196-8
“The inclusion criteria of all of the studies were a diagnosis of mild to moderate CTS through NCVs diagnosis and physical examination according to the different criteria [10, 14, 21, 25, 28-35].”
Since there are only five studies among the included 12 papers reporting results of nerve conduction study, please clarify how the authors defined mild to moderate CTS according to results of physical examination.
3. In the footnote of Tables 5 and 6, it is confusing regarding the statement in the notes.
“Notes: 1Excluded since MCV, SCV and motor latency were not recorded separately. 2Exlcuded from analysis since study did not record “motor conduction velocity”. NM = neurodynamic modulation.”
Table 5 showed reference 10 did not report SCV data and Table 6 showed it was excluded because of not recording MCV. In Table 7, reference 10 did not report motor latency, either. I wonder why the authors included reference 10 in the above three tables since it did not report any of the three NCS outcomes.
Did any of the 5 studies report distal sensory latency of the median nerve? I believe it is more sensitive than the above three outcomes and meta-analysis of its changes is recommended.
4. Please compare the results of this study with previous meta-analysis studies and elaborate the new contribution of this study.
5. Since there are 5 studies conducted by the same authors group, please conduct funnel plots to evaluate potential publication bias.
6. High heterogeneity might be caused by high varieties of different combinations of both intervention and control groups. Some intervention groups used neurodynamic techniques combined with carpal bone mobilization or electrotherapy, and the control groups included active control or no intervention. Network meta-analysis might be a better method to solve this problem.
Author Response
Comment 1: Although the authors made some discussion regarding the discrepancies of neuromodulation (NM) implemented in the included 12 studies, it is recommended to make a definition of NM technique and explain their applying methods (e.g., gliding and sliding maneuvers) in the Introduction section.
Author response: This has now been added in Lines 71-74.
Comment 2: Lines 196-8 - “The inclusion criteria of all of the studies were a diagnosis of mild to moderate CTS through NCVs diagnosis and physical examination according to the different criteria [10, 14, 21, 25, 28-35].”
Since there are only five studies among the included 12 papers reporting results of nerve conduction study, please clarify how the authors defined mild to moderate CTS according to results of physical examination.
Author response: We have deleted this statement as it is incorrect and not appropriate here (now from Line 207).
Comment 3: In the footnote of Tables 5 and 6, it is confusing regarding the statement in the notes. “Notes: 1Excluded since MCV, SCV and motor latency were not recorded separately. 2Exlcuded from analysis since study did not record “motor conduction velocity”. NM = neurodynamic modulation.”
Table 5 showed reference 10 did not report SCV data and Table 6 showed it was excluded because of not recording MCV. In Table 7, reference 10 did not report motor latency, either. I wonder why the authors included reference 10 in the above three tables since it did not report any of the three NCS outcomes.
Author response: Studies where data is not available have been deleted and hence the footnotes have also been deleted.
Did any of the 5 studies report distal sensory latency of the median nerve? I believe it is more sensitive than the above three outcomes and meta-analysis of its changes is recommended.
Author response: Unfortunately, only one study reported distal sensory latency whilst 2 studies reported distal motor latency and hence a meta-analysis is not possible with so few studies.
Comment 4: Please compare the results of this study with previous meta-analysis studies and elaborate the new contribution of this study.
Author response: This has now been included as a new paragraph, Lines 419-430.
Comment 5: Since there are 5 studies conducted by the same authors group, please conduct funnel plots to evaluate potential publication bias.
Author response: Although there are 5 studies from the same author group, the maximum number of studies in the outcomes assessed were 3 of the 5 studies. A funnel plot did reveal that the studies were low precision and likely to be bias since they were not distributed in a funnel shape. However, we do not think that adding funnel is useful. However, an example plot is shown below for the reviewer’s benefit.
Funnel plot for 3 studies from the same author group to demonstrate possible publication bias in BCTQ combined (NM vs other therapy).
Comment 6: High heterogeneity might be caused by high varieties of different combinations of both intervention and control groups. Some intervention groups used neurodynamic techniques combined with carpal bone mobilization or electrotherapy, and the control groups included active control or no intervention. Network meta-analysis might be a better method to solve this problem.
Author response: We agree that this is a great suggestion however the intent of our systematic review was to answer if NM was superior than other techniques including no therapy. This is so that it encompasses differences in standard therapies across the world. Therefore, a network meta-analysis is no appropriate here.
Round 2
Reviewer 2 Report
Thanks for the authors' effort. Most issues were resolved after major revision.